# A Multi-Agent Approach Used to Predict Long-Term Glucose Oscillation in Individuals with Type 1 Diabetes

João Paulo Aragão Pereira [1,*], Anarosa Alves Franco Brandão [1,*], Joyce da Silva Bevilacqua [2] and Maria Lucia Cardillo Côrrea-Giannella [3]

1   Computing Engineering Department, Escola Politécnica, Universidade de São Paulo,
    São Paulo 05508-010, Brazil
2   Applied Mathematics Department, Instituto de Matemática e Estatísitica, Universidade de São Paulo,
    São Paulo 05508-090, Brazil
3   Laboratory of Carbohydrates and Radioimmunoassays (LIM-18), Hospital das Clínicas (HCFMUSP),
    Faculdade de Medicina, Universidade de São Paulo, São Paulo 05403-900, Brazil
*   Correspondence: raijoma@usp.br (J.P.A.P.); anarosa.brandao@usp.br (A.A.F.B.)

**Featured Application: This work provides a support and decision system that can support endocrinologists in telemedicine, hospitals, insurance companies and analysis laboratories by being the basis for applications. In addition, it can support end users as a recommendation system and glucose oscillation prediction. Moreover, it can be embedded in insulin pumps (devices or software).**

**Abstract:** The glucose–insulin regulatory system and its glucose oscillations is a recurring theme in the literature because of its impact on human lives, mostly the ones affected by diabetes mellitus. Several approaches have been proposed, from mathematical to data-based models, with the aim of modeling the glucose oscillation curve. Having such a curve, it is possible to predict when to inject insulin in type 1 diabetes (T1D) individuals. However, the literature presents prediction horizons of no longer than 6 h, which could be a problem considering their sleeping time. This work presents Tesseratus, a model that adopts a multi-agent approach used to combine machine learning and mathematical modeling to predict the glucose oscillation for up to 8 h. Tesseratus can support endocrinologists and provide personalized recommendations for T1D individuals to keep their glucose concentration in the ideal range. It brings pioneering results in an experiment with seven real T1D individuals. Using the Parkes error grid as an evaluation metric, it can be depicted that 93.7% of measurements fall in zones A and B during the night period with MAE 27.77 mg/dL. It is our claim that Tesseratus will be a reference for the classification of a glucose prediction model, supporting the mitigation of long-term complications in the T1D individuals.

**Keywords:** glucose oscillation; prediction; multi-agent; type 1 diabetes; personalized; recommendation

## 1. Introduction

Diabetes mellitus is a syndrome characterized by hyperglycemia resulting from defects in insulin secretion, associated or not with resistance to the action of this hormone [1]. Diabetes mellitus affects people all around the world, and it is an important health challenge for the 21st century. Type 1 diabetes mellitus (T1D) is the type in which the pancreas produces little or no insulin [1], causing glucose concentration control problems. Such problems must be treated with insulin injection and frequently causes long-term complications. Currently, more than 1.1 million children and teenagers have T1D around the world [2].

There are several challenges faced by researchers and health professionals, mostly related to developing means to provide treatments that bring better results in terms of short-term glucose control and risks reduction in developing long-term complications. Efforts to address these challenges come from academia to the pharmaceutical industry.

From the theoretical and technological point of view, several models were proposed to support the avoidance of hypoglycemic or hyperglycemic conditions in T1D individuals. Moreover, there are models that mimic the dynamic exchange of information between the components of the human glucose–insulin regulatory system (HGIRS) [3]. Nevertheless, to the best of our knowledge, they are still unable to predict the glucose oscillation curve for prediction horizons greater than six hours.

This paper presents Tesseratus, a hybrid model that adopts a multi-agent system (MAS) [4] approach to combine math and data-driven techniques to predict the glucose oscillation for up to eight hours for individuals with T1D. As the *tesseract* has four dimensions [5], Tesseratus has four main agents, namely (i) the `Recommender`; (ii) the `Predictor`; (iii) the `Math` and (iv) the `ML` agents. In addition, the model also has reactive agents to receive external data from the T1D individual and to monitor and control inputs and outputs along the system execution.

Tesseratus predicts the behavior of the glucose oscillation by taking into account basal and fast-acting insulin infusion; current glucose concentration; food (carbohydrate, protein and fat) and alcohol intake; and physical activity. Having such information, the model is able to present the predicted glucose curve for a prediction horizon (PH) for up to four hours during the day and up to eight hours during the night.

This paper is organized as follows: Section 2 presents a brief description of the human glucose–insulin regulatory system and its associated mathematical model. Section 3 presents some related work and Section 4 presents the Tesseratus model itself, its architecture and functioning. After, we present the experimental results in Section 5 and the associated discussion in Section 6. Finally, the paper is concluded in Section 7.

## 2. Human Glucose–Insulin Regulatory System

The human glucose–insulin regulatory system (HGIRS) is part of the human endocrine system [6] and comprises two main hormones, insulin and glucagon, produced and released, respectively, by $\beta$ and $\alpha$-cells of the pancreatic islets. These two hormones, which exert opposing effects, act in concert to maintain blood glucose (BG) in a narrow range. During the fasting state, glucagon stimulates hepatic glucose production (HGP) in order to prevent hypoglycemia, while insulin is secreted at levels sufficient to constrain HGP and to maintain BG concentration at approximately 90 mg/dL (basal secretion). After meals, the increase in BG concentration stimulates insulin secretion (meal-related secretion), suppressing HGP and stimulating glucose uptake by insulin-sensitive tissues such as muscle and adipose tissue, eventually restoring normoglycemia. In this research, due to modeling limitations, we do not consider other hormones such as cortisol and growth hormones, which exert direct and indirect effects on glucose metabolism.

Insulin secretion is complex and glucose is the most potent stimulant of insulin release. After eight to ten minutes of food ingestion, the insulin concentration increases, reaching a peak in 30 to 45 min, and then rapidly decreases to baseline values in 90 to 120 min [7]. Insulin is a physiological suppressor of glucagon release; thus, glucagon secretion is low in the postprandial period. On the other hand, glucagon is released during fasting, when BG is in the normal range and insulin concentration is low [7].

In the case of T1D, the autoimmune destruction of pancreatic $\beta$-cells prevents insulin secretion. Thus, T1D individuals depend on exogenous insulin administration to mimic the physiological secretion of this hormone, i.e., basal and meal bolus insulin. To maintain the BG concentration to as close as possible to the normal range, it is also necessary to measure BG and to count the amount of macronutrients (especially carbohydrates) before every meal to calculate the bolus insulin dose, which must match the total carbohydrate content of that meal and also correct occasional hyperglycemias.

Optimal glycemic control is crucial to avoid the complications associated with chronic hyperglycemia. However, the procedures described above are relatively complex and are influenced by numerous other factors, such as the type and intensity of physical activity, as well as stress, among others, which impair glycemic control, contributing to the occurrence

of episodes of hyper and hypoglycemia. The literature presents several predictive models for glucose oscillation, such as [8–11], the last one having a night prediction of up to 6 h. However, we could not find any model that predicts glucose oscillation in a continuous and personalized way.

Quantitative methods used to model the metabolic physiology of T1D individuals are usually based on ordinary differential equations (ODE), and are also called compartment models [12]. We propose an extension to the model proposed by Kissler et al., 2014 [3] presented in Equations (1) and (2).

$$G'(t) = G_{in} + f_1(I(t - \tau_2)) - f_2(G(t) \\ -\gamma[1 + s.(m - m_b)].(f_3(G(t)).f_4(I(t)) \tag{1}$$

$$I'(t) = I_{in} + \beta f_5(G(t - \tau_1)) - \frac{V_{MAX}I(t)}{K_M + I(t)} \tag{2}$$

In the glucose compartment $G'(t)$, $f_1$ describes the hepatic glucose production (HGP); $f_2$ describes the central nervous system glucose utilization; $f_3$ describes the muscle/fat glucose utilization; $f_4$ describes the muscle/fat insulin uptake; and $f_5$ describes the pancreatic insulin production. The parameters semantics are given in Table 1.

**Table 1.** Parameters of the glucose and insulin compartment models.

| Symbol | Description |
| --- | --- |
| $I_{in}$ | Insulin infusion rate |
| $G_{in}$ | Glucose intake rate |
| $\beta$ | Relative pancreatic $\beta$-cell function |
| $\gamma$ | Relative insulin sensitivity |
| $s$ | Rate of insulin sensitivity increase per minute of exercise |
| $m$ | Daily minutes of physical activity |
| $m_b$ | Baseline minutes of physical activity |
| $V_{max}$ | Maximum insulin clearance rate |
| $K_M$ | Enzyme's half-saturation value |

Our extension considers the approach proposed by Schindelboeck et al., 2016 [13] for describing $f_1$ in compartment $G'(t)$, in order to consider alcohol ingestion and replace $\gamma$ proportionally by $eGDR$—the estimated glucose disposal rate—following the approach of Epstein et al., 2013 [14]. Nevertheless, we completely exchanged compartment $I'(t)$ by building piecewise polynomial equations from the pharmacokinetics data of four types of insulin. This mathematical modeling is the core of our `Math` agent and is described in the following.

### 2.1. A Mathematical Model for HGIRS

The glucose equation $G'(t)$ (glycemic value as a function of time) should be calculated and is directly related to the amount and type of macronutrients ingested, as well as the time ($\Delta t_{ex}$–measured in minutes) and intensity ($VO_2$—maximum volume of oxygen consumed) of physical exercise [15,16] in accordance with [17].

$G_{in}$ and $I_{in}$ values refer to the rate of glucose intake and insulin infusion, respectively. $G_{in}$ is measured in mg/dL.min, varying in the interval $[0, 1.08]$. The insulin equation $I'(t)$ (insulin concentration value as a function of time) and the value of $I_0$ (insulin concentration at $t(0)$) come from the pharmacokinetics equations of each type of insulin (Section 2.1.2) selected by each individual.

Thus, having these two equations modeled, we fed our `Math` agent with them in order to start the labeling of our dataset, as well as to support the continuous learning in Tesseratus.

### 2.1.1. The Glucose Compartment Equation

The description of how the glucose compartment is modeled by using a similar approach of Equation (1), extending $f_1$ and reusing $f_2$, $f_3$ and $f_4$. In addition, we rename the physical exercise contribution to the model ($f_{ex} = [1 + s(\Delta t_{ex} - \overline{\Delta t_{ex}})]$). Therefore, it is represented by

$$
G'(t) = \overbrace{(G_{in} + f_1(I(t - \tau_1)))}^{\text{glucose production}} - \\
\underbrace{(f_2(G(t)) + \gamma f_{ex}.(f_3(G(t)).f_4(I(t)))}_{\text{glucose consumption}}
\tag{3}
$$

Our extension in $f_1$ considers the fact that there are two sources of glucose production: the hepatic glucose production (HGP) [3] and the glucose yielded from the metabolism of ingested macronutrients. In this case, glucagon exerts control over the liver and causes it to dispense glucose, with a slight delay (given by $\tau_1$) of between 15 and 20 min [18]. In order to allow for personalization, we defined $f_1$ considering or not considering alcohol ingestion, following the understanding of Schinfelboeck et al., 2016 [13]. Both Equations (4) and (5) use the reference values proposed by [19–21]. Then, $HGP_{max}$ is 180 mg/min, $\alpha$ is 0.29 L/mU, $V_{pla}$ is 3 L and $C_5$ is 26 µU/L. Here, $HGP_{max}$ stands for hepatic glucose production, $\alpha$ for hepatic sensitivity to changes in insulin, $V_{pla}$ for the volume of plasma in the body, $C_5$ for the insulin concentration at which the liver is most efficient and $A_g(t)$ for alcohol ingestion.

$$
f_1(I(t - \tau_1)) = \frac{HGP_{max}.(1 - A_g(t))}{(1 + \exp(\alpha(\frac{I(t)}{V_{pla}}) - C_5))}
\tag{4}
$$

or

$$
f_1(I(t - \tau_1)) = \frac{HGP_{max}}{(1 + \exp(\alpha(\frac{I(t)}{V_{pla}}) - C_5))}
\tag{5}
$$

### 2.1.2. The Exogenous Insulin Equation

Our approach adopted the pharmacokinetics data of four types of insulin (glargine [22], degludec [23], lispro [24] and aspart [25]) to support this modeling based on the approximation of polynomial functions. Since industry information is not enough for producing a viable approximation, we combined them with information from a dataset of seven real Brazilian volunteers that use such insulins to derive the polinomyal functions related to each of them. The dataset is described in Section 5.

For instance, for glargine, an insulin of slow action, we obtained values from [26–30], as well as the FDA report [22] and the industry representative information [31]. On the other hand, for lispro, an insulin of fast action, we obtained values from [32–34]. The insulin lispro report produced by FDA [35] was used to confirm the time-of-action and pharmacokinetic information. Having such values, we were able to build a polynomial function $p(t)$ that provides an approximation for the exogenous insulin compartment. Each point of $p(t)$ represents the concentration of insulin prescribed at a given time ($t$), considering the parameters of the T1D individual.

Figure 1 presents an example of a polynomial curve $p(t)$ for glargine to cover a 24 h period of time. It can be observed that $p(t)$ is a piecewise combination of polynomials of degrees four, two and three, as described in Equation (6).

$$p(t) = \begin{cases} -0.0259t^4 + 0.4255t^3 - 2.5787t^2 + 7.213t + 9.5966, & 0 \leq t \leq 6 \\ 0.105t^2 - 2t + 26.655, & 6 < t \leq 8 \\ -0.0996t^4 + 3.8587t^3 - 57.701t^2 + 379.78t - 910.29, & 8 < t \leq 12 \\ -0.0808t^3 + 3.5536t^2 - 52.254t + 269.54, & 12 < t \leq 16 \\ -0.011t^3 + 0.6492t^2 - 13.24t + 102.98, & 16 < t \leq 24 \end{cases} \quad (6)$$

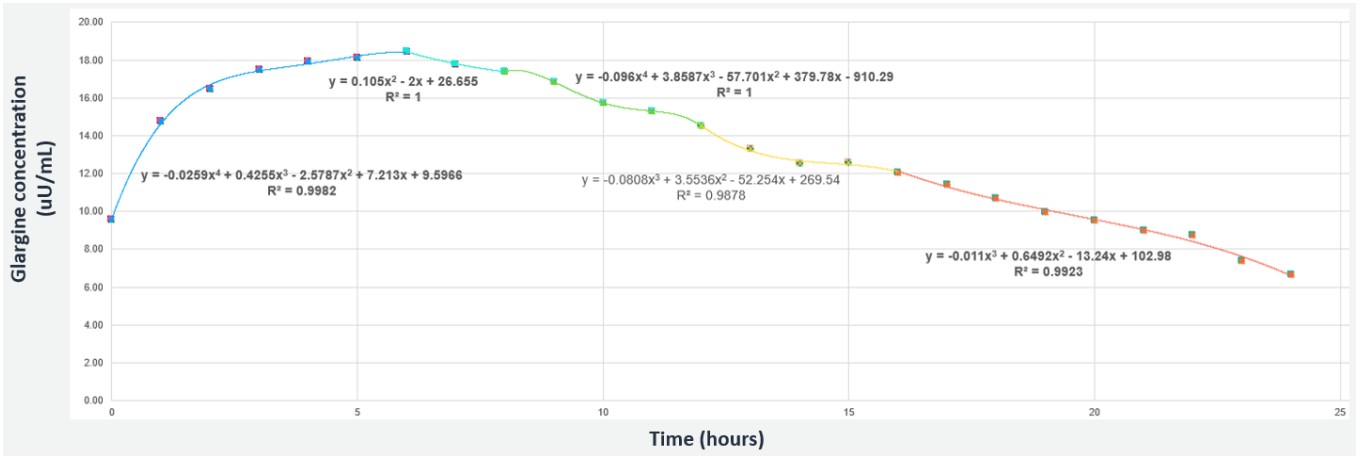

**Figure 1.** The piecewise polynomial for Glargine insulin (slow-acting) considering 0.36 U/kg concentration.

Figure 2 presents an example of a polynomial curve $p(t)$ for lispro that covers a period of time of 300 min. It can be observed that $p(t)$ is a piecewise combination of polynomials of degrees three and two, as described in Equation (7).

$$p(t) = \begin{cases} -0.0015t^3 + 0.0752t^2 + 0.0473t + 0.082, & 0 \leq t \leq 35 \\ -0.0022t^2 + 0.2445t + 23.638, & 35 < t \leq 45 \\ -0.0097t^2 + 0.9403t + 7.5778, & 45 < t \leq 60 \\ 0.00002t^3 - 0.0059t^2 + 0.3791t + 23.134, & 60 < t \leq 120 \\ 0.0004t^2 - 0.2612t + 42.8774, & 120 < t \leq 220 \\ 0.000006t^2 - 0.0822t + 22.163, & 220 < t \leq 300 \end{cases} \quad (7)$$

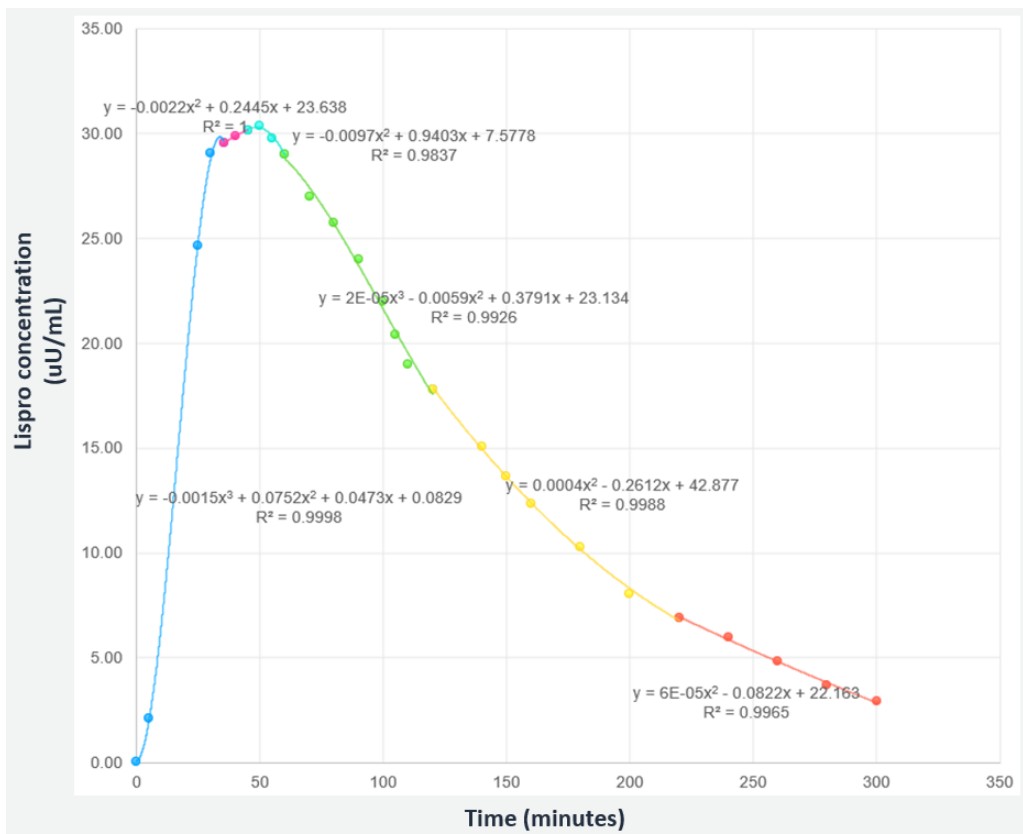

**Figure 2.** The piecewise polynomial for Lispro insulin (fast-acting) considering 0.16 U/kg concentration.

## 3. Related Work

This section presents some models that provide hybrid solutions for the predictive modeling of glucose oscillation for T1D. Hybrid solutions usually combine mathematical modeling with data-driven techniques. In a literature search, Refs. [8–10,36] presented hybrid predictive models with PHs from 90 to 120 min. Nevertheless, for [16,37–46], the prediction horizon (PH) varies from 30 to 60 min.

In fact, Refs. [37,41,42] adopt neural network approaches that yield a PH of 60 min when dealing with nine, six and ten real patients, respectively; Ref. [38] adopts a combination of a compartment model and self-organizing map with twelve real patients, achieving a PH of 60 min; Refs. [39,40] adopt a support vector machine approach tested with six and five real patients, respectively, both achieving a PH of 60 min. Moreover, Refs. [43–46] achieved a PH of 30 min by adopting Bayesian inference, fuzzy logic, neural networks and a Kalman filter, respectively.

For long-term PH, we can state that Georga and colleagues et al., 2013 [8] adopted a multivariate regression approach to derive a predictive model for subcutaneous glucose concentration prediction in T1D individuals. The method was evaluated with a dataset composed of twenty-seven real T1D individuals and presented average prediction mean square errors of 5.21 mg/dL for 15 min, 6.03 mg/dL for 30 min, 7.14 mg/dL for 60 min and 7.62 mg/dL for 120 min PHs. Liu and colleagues et al., 2019 [9] presented a glucose forecasting algorithm suited for long-term PHs. The algorithm is based on compartmental models for the HGIRS. It was evaluated with clinical data of ten real T1D individuals. For a 120 min PH, there was an improvement of 18.8% in the prediction accuracy measured with the root mean square error (RMSE), 17.9% in the A-region of error grid analysis (EGA) and 80.9% in the hypoglycaemia prediction calculated by the Matthews correlation coefficient. Cescon, Johansson and Renard et al., 2015 [10] presented a subspace-based linear multi-step predictor as a predictive model for short-term glucose oscillation. The model was evaluated with seven real T1D individuals and obtained a prediction error standard deviation of 58.06 mg/dL at 120 min. Contreras and colleagues et al., 2018 [36] combined physiological



models for HGIRS with grammatical evolution as a search-based technique to design a predictive model for short-term glucose oscillation. Considering the CEG, they achieved more than 96% of results falling inside regions A and B for 90 min.

Since there are T1D individuals that feed at intervals greater than two hours, their results are not helpful for them. The proposal of the Tesseratus model is to extract the best features of each approach (physiology, data and multi-agent) in order to reach a longer PH at personalized intervals of four hours in the full daytime period and eight hours in the night period, keeping the MAE below 28 mg/dL in both periods and for all PHs. A discussion of the results is in Section 5.

## 4. Tesseratus Model

Tesseratus adopts a multi-agent approach to define a hybrid model to predict the glucose oscillation for up to four hours during daytime and for up to eight hours at the night period. It is a hybrid model because agents are defined using both compartment models (described in Section 2.1) and data-driven techniques, such as machine learning (ML). Tesseratus has two types of agents: reactive agents and intelligent agents. Nevertheless, we decomposed the problem in such a way that each agent is responsible for a portion of the phenomenon [47], which means that there are several agents of each type. The reactive agents are responsible for collecting data and feeding intelligent agents with the collected data or for monitoring data, error and ODE parameters. Intelligent agents are responsible for using data to predict the glucose oscillation. The intelligent agents are: the `Recommender` agent, the `Predictor` agent, the `ML` agent and the `Math` agent.

The `Recommender` agent has a knowledge base composed of labeled glucose oscillation prediction curves (within the range or out of range—following a semaphore metaphor) and is responsible for finding and labeling suitable curves for received data and delivering them to the user. The `Predictor` agent has a knowledge base of labeled predicted values and actions and is responsible for asking the `Math` and `ML` agents for information to populate its knowledge base and for providing glucose oscillation prediction curves to the `Recommender` agent. The `Math` agent has the math modeling presented in Section 2.1 as the core for the generation of glucose oscillation prediction curves, and the `ML` agent learns the glucose oscillation prediction curves from the combination of received data provided by reactive agents and by the `Math` agent. Communication is bidirectional between intelligent agents. The `Predictor` agent communicates with all intelligent agents, and the `ML` agent communicates with the `Predictor` and `Math` agents. Reactive agents only send messages to intelligent agents. Having received these messages, intelligent agents act accordingly to recommend a glucose oscillation prediction curve or to adapt the recommendation given the monitored context (prediction error above a threshold, the need for updating ODE parameters or non-conformance of collected data as expected).

Tesseratus innovates in the problem solution by adopting a learning policy that considers both information from the individuals collected data and from the mathematical modeling. Therefore, at the very beginning, we defined a set-up step for knowledge acquisition and data labeling. In this step, the `Math` agent generates glucose values and associated labeling, providing information to accelerate the learning using the generated glucose values and associate labeling, and the `ML` agent reuses knowledge from the `Math` agent. All intelligent agents knowledge bases are built in a continuous reinforcement learning cycle that begins with the `Math` agent knowledge combined with a reward policy based on the semaphore metaphor. Data are labeled `green`, `yellow` or `red` depending on the glucose concentration and the absolute error, both within pre-established thresholds. The reader may observe in Table 2 that there are two rewards labeled as `green` and two other rewards labeled as `red` because excellent and normal glucose levels associated with an absolute error smaller than 30 mg/dL are labeled as `green`, depending on the PH; and hyperglycemia and hypoglycemia levels are labeled as `red`, as well as an absolute error bigger than 30 mg/dL. For the `red` label, the reward should be considered if at least one of

the situations occur. An acceptable glucose concentration is labeled as `yellow`. By absolute error, we mean the error between the measured and predicted glucose values.

**Table 2.** Tesseratus reward policy.

| State | Reward |
| --- | --- |
| Green | 10 |
| Green | 8 |
| Yellow | 0 |
| Red | −5 |
| Red | −10 |

The architecture of Tesseratus is presented in Figure 3, where numbers are adopted to support its explanation flow.

1. Reactive agents collect data from continuous glucose monitors (CGM), voice or manually, and send them to the `Recommender` agent;
2. The `Recommender` agent receives data, associates them with their time frame, creating a tuple $< data, time >$, and checks its knowledge base (KB) if there are actions to be taken related to them;
3. If yes, the predicted oscillation curve is labeled in the ideal range (80–120 mg/dL during fasting, and up to 160 mg/dL in postprandial periods), stored in the KB as $< time, label, curve >$, and sends it to the user;
4. If not, the `Recommender` agent requests information about prediction curves to the `Predictor` agent;
5. The `Predictor agent` checks its KB to see if there is a suitable prediction curve. If not, it propagates the request to the `ML` and `Math` agents;
6. The `ML` agent and `Math` agent, at a given time frame, store the prediction values in their KB and return the value linked with the prediction calculation to the `Predictor` agent;
7. The `Predictor` agent analyzes the value received and, if it is a value that is in the ideal range, sends a return message to the `Recommender` agent; otherwise, it requests more options for the `ML` and `Math` agents. At this point, the `Predictor` agent stores the input and output in its KB with a specific timestamp;
8. The `Recommender` agent could send recommendations to the environment from its base of action , or simply send the predicted oscillation curve with an ideal glucose label achieved;
9. The best glucose value is sent to the environment, as well as hypothetical complementary recommendations;
10. The actions and knowledge at that specific time frame are stored in the `Recommender` agent's KB.

*4.1. Tesseratus Implementation*

All agents were implemented in a serverless architecture based on microservices [48]. The multi-agent system (MAS) follows an event-based approach and runs on a public cloud platform [49]. The KB of each intelligent agent uses a non-relational, non-server management key-value database, with flexibility for unstructured data, and also for being horizontally scalable.

A Smart Python Agent Development Environment (SPADE) [50] was used as the multiagent platform. Communication among agents is based on instant messaging, an interesting feature that allows for presence notification, enabling the system to know the current state of the agents in real-time. SPADE agents are based on behaviors, and were extended to create the hybrid agents with a Belief–Desire–Intention (BDI) [51] layer.

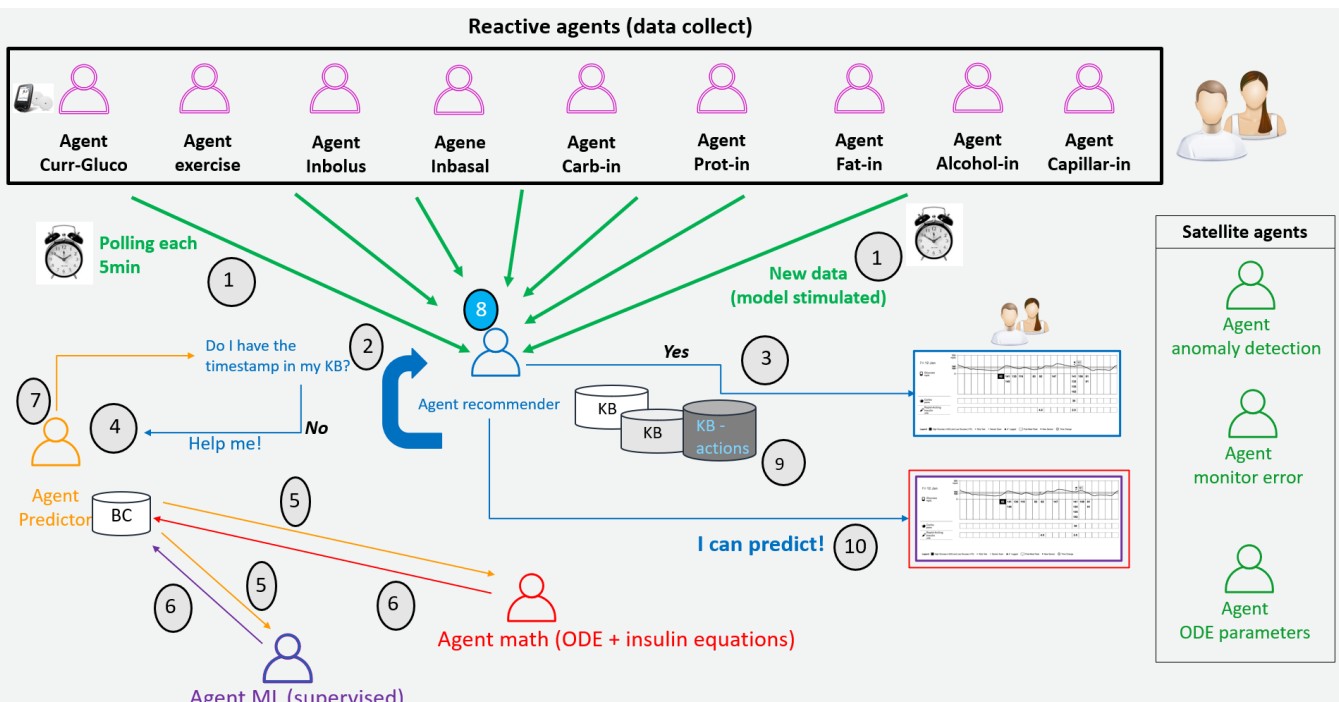

**Figure 3.** Tesseratus architecture.

*4.2. Tesseratus Set Up*

In order to prepare the model to be used, there is a setting-up step for knowledge acquisition. In this step, intelligent agents use data provided by the T1D individual during a week to create their associated KB. As aforementioned, the `Math` agent accelerates the `ML` agent learning by providing information regarding glucose prediction values based on the HGIRS mathematical modeling presented in Section 2.1. In addition, the `Math` also uses the pre-defined semaphore to label generated values as `green`, `yellow` and `red`. Nevertheless, `ML` agent uses the labeled data to learn in a supervised learning process [4]. This step adopts an approach similar to active learning. Figure 4 depicts this learning flow, which is described next.

1.  Reactive agents collect environmental information via sensors (every 5 min from GCM), by voice or manually;
2.  The `Recommender` agent receives data and asks the `Predictor` agent to calculate and generate the glucose curve;
3.  The `Predictor` agent asks the `ML` agent to create a timestamp and generate the glucose curve;
4.  The `ML` agent asks the `Math` agent to label the generated value based on the predefined semaphore;
5.  The `Math` agent forwards the labeling payload to the `ML` agent;
6.  The `ML` agent sends to the `Predictor` agent the timestamp, glucose curve, predictions and label based on the data received;
7.  The `Predictor` agent receives the message and forwards it to the `Recommender` agent;
8.  The `Predictor` agent receives the new information and classifies it in each KB according to the received label (`green`, `yellow` and `red`).

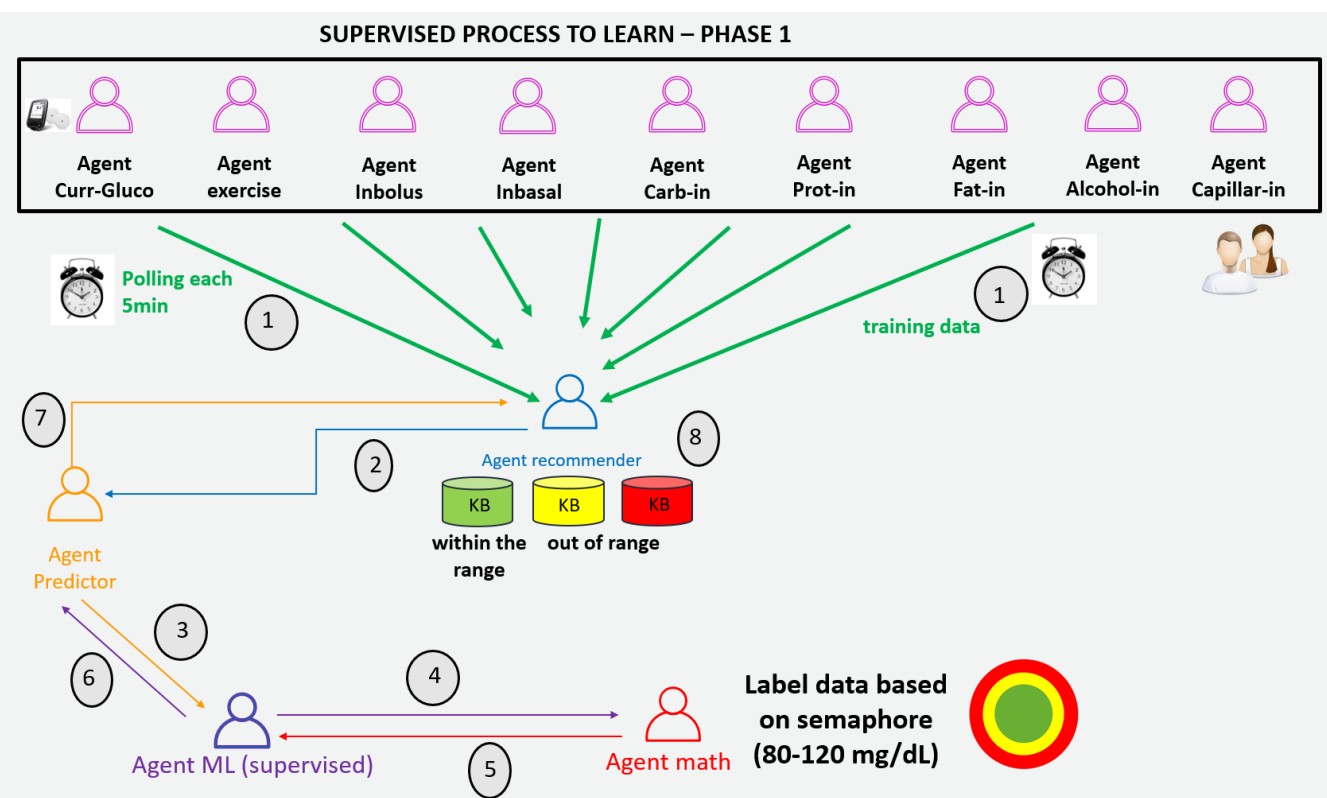

**Figure 4.** Tesseratus set up: acquiring knowledge.

*4.3. Tesseratus Functioning*

After the set up, Tesseratus starts its functioning in a continuous learning cycle based on reinforcement learning, with receipt of new data provided by the reactive agents in order to carry out the validation of values.

The `Math` and `ML` agents generate knowledge for the `Recommender` and `Predictor` agents. In addition, they also take advantage of information provided by the other reactive agents in order to update and/or correct the values of the parameters used in the ODE, which are the core of the `Math` agent. Moreover, we consider the time series associated with the green labeled data to adopt the sliding window approach [52] and achieve different PHs, from 15 min to 8 h. For instance, since the glucose concentration dataset is measured every five minutes, the `ML` agent used the last two hours of historical data on carbohydrates and five hours on bolus insulin to predict it (see green and black dashed left-right arrows of Figure 5). These time slots were chosen based on the duration of carbohydrate metabolism and the average time of fast-acting insulin action in the human body, respectively. In Figure 5, insulin values (in mU/mL) of the five-hour sliding window are represented by the black dashed left-right arrow, whereas carbohydrate values (in grams) of the two-hour sliding window are represented by the green dashed left-right arrow. They were used to predict the oscillation glucose for up to eight hours (blue dashed right arrow).

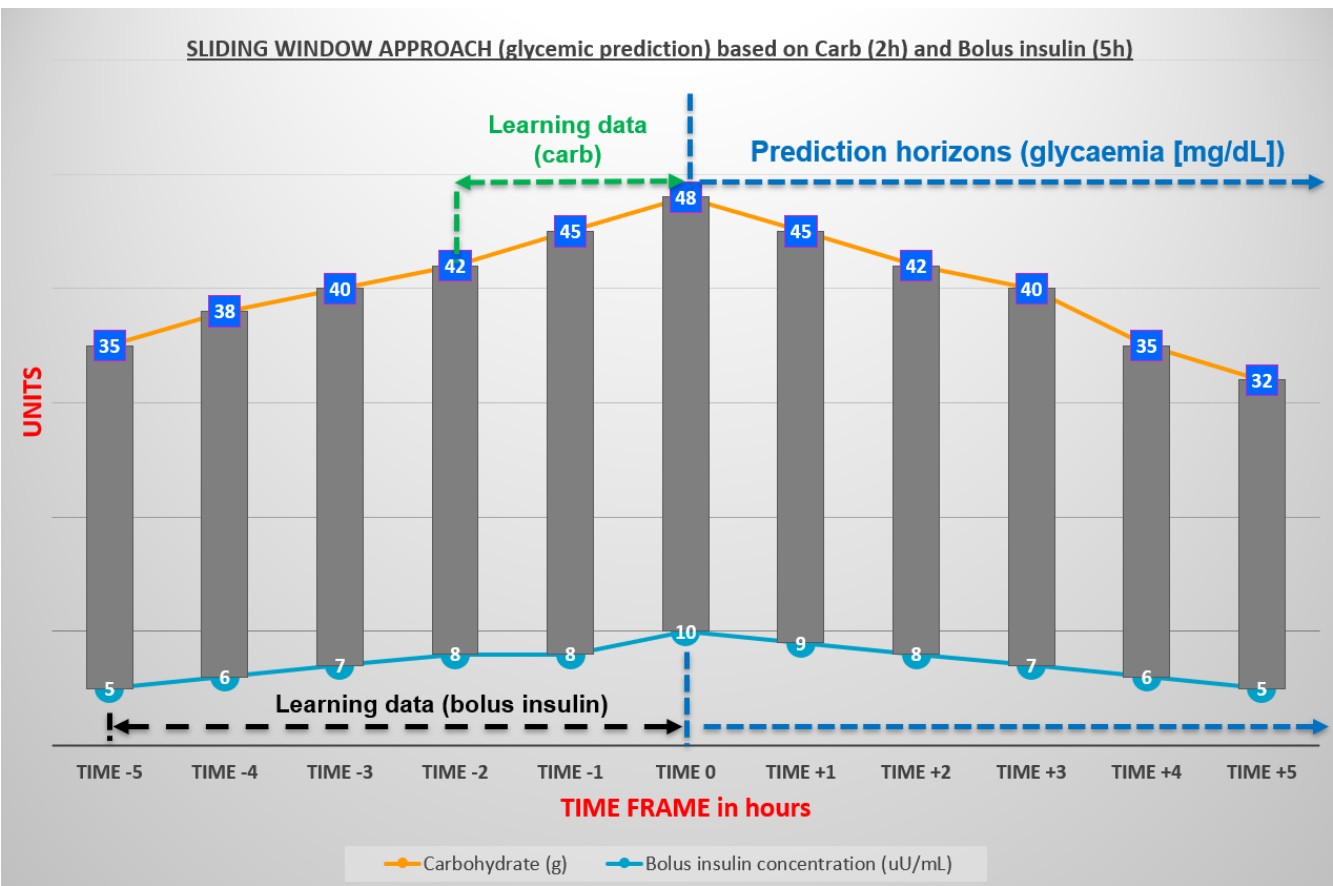

**Figure 5.** Sliding window approach for time series: carbohydrates and insulin.

## 5. Results

Tesseratus was implemented as a Python [53] prototype (version 3.8). Its setup was conducted considering data collected from seven T1D Brazilian volunteers during a week. The volunteers profiles are described in Table 3, where their ID, sex, age, abdominal circumference (AC) and body mass (BM) are given. In addition, Table 4 presents the time of data collection, the origin of glucose values and the type of insulin, with a concentration of 100 units per mL (100U/mL). Other data collected from the T1D individuals were the HbA1c (hemoglobin A1c) value and whether they were hypertensive (1) or not (0) in order to calculate the eGDR (mg/(kg min)) of each individual (Table 5). The ID is only used to identify the individual.

**Table 3.** Characteristics of each individual.

| ID | Sex (M/F) | Age (years) | AC (cm) | BM (kg) |
|---------|-----------|-------------|---------|---------|
| R-BRA01 | F | 36 | 79 | 57 |
| R-BRA02 | M | 50 | 115 | 90 |
| R-BRA03 | M | 28 | 91 | 70 |
| R-BRA04 | M | 45 | 132 | 120 |
| R-BRA05 | M | 39 | 93 | 83 |
| R-BRA06 | M | 39 | 91 | 78 |
| R-BRA07 | F | 65 | 90 | 68 |

**Table 4.** Satellite characteristics used to collect data.

| ID | Days | Source | Insulin |
|---|---|---|---|
| R-BRA01 | 14 | FreeStyle Libre® | aspart + degludec |
| R-BRA02 | 21 | Minimed 640G® | lispro |
| R-BRA03 | 14 | FreeStyle Libre® | aspart + degludec |
| R-BRA04 | 21 | Minimed 640G® | lispro |
| R-BRA05 | 14 | FreeStyle Libre® | aspart + glargine |
| R-BRA06 | 21 | Paradigm VEO 754® | lispro |
| R-BRA07 | 14 | FreeStyle Libre® | lispro + degludec |

**Table 5.** Information used to calculate estimated glucose disposal rate (eGDR).

| ID | Hypertensive | HbA1c (%) | eGDR |
|---|---|---|---|
| R-BRA01 | 0 | 5 | 11.7 |
| R-BRA02 | 1 | 6 | 4.51 |
| R-BRA03 | 0 | 6.3 | 9.91 |
| R-BRA04 | 1 | 5.6 | 3.2 |
| R-BRA05 | 0 | 8 | 8.8 |
| R-BRA06 | 0 | 6 | 10 |
| R-BRA07 | 0 | 7.5 | 9.34 |

An example of how it is possible to analyze the behavior of each individual is represented in Figure 6 for volunteer R-BRA07. Her profile is characterized in Tables 3–5: R-BRA07 is non-hypertensive, and her hemoglobin A1c is 7.5%, yielding an eGDR of 9.34 (mg/kg min). The glucose rate of ascent and descent was collected every 30 min, between midnight and 8am, during 14 days. These data were compared with the Tesseratus prediction ones and further plotted in a Parkes error grid [54] (Section 5.1). It is possible to notice that there is a tendency for the glucose concentration to fall in the early hours of the morning, but, for example, there is a rise every day from five in the morning. The glucose oscillation is quite peculiar and can be affected by the dawn phenomenon, which is characterized by hyperglycemia during early morning [55]. Another factor that influences the continuity of the glucose concentration increase is having breakfast almost every day at six o'clock with a carbohydrate intake.

### 5.1. Testing and Validation

Through the natural competition established between the `Math` and `ML` agents, with their respective strategies, it was possible to establish the best result between them and practice a continuous flow of active reinforcement learning using historical data from seven volunteers for up to 21 days. The best result is always closer to the ideal glucose range: 80–120 mg/dL in fasting and up to 160 mg/dL in the postprandial period.

Daytime and night-time windows were personalized for each individual (Table 6), with the addition of information related to their usual sleeping hours (night-time) and active hours (daytime). Such personalization is needed to decide whether Tesseratus must be fed by a new external stimulus. In addition, our prediction horizon for the night-time is eight hours at most, and individuals with night-time windows greater than that must have their prediction horizon updated.

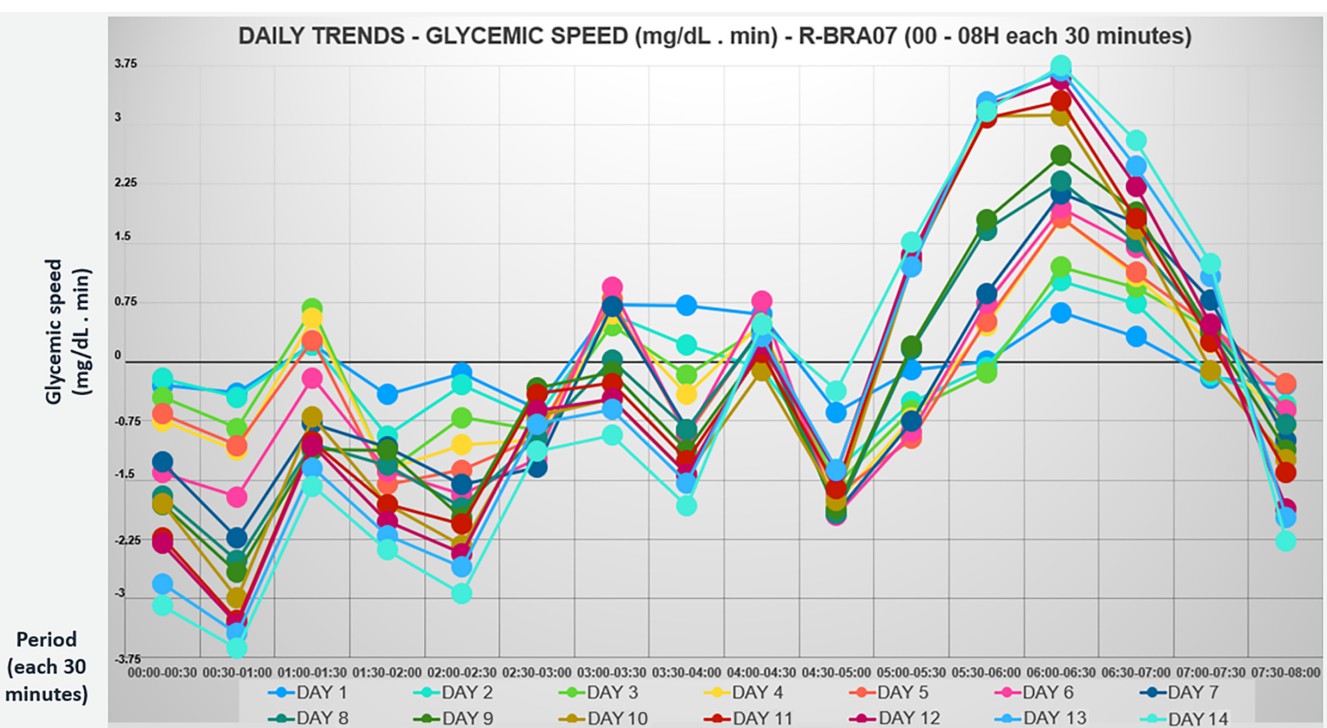

**Figure 6.** Ascent and descent glucose rate from individual R-BRA07 during 14 days along nighttime (from midnight to eight am) considering time intervals of 30 min.

**Table 6.** Volunteers' nighttime and daytime windows.

| ID | Nighttime | Daytime |
|---|---|---|
| R-BRA01 | 10:41 p.m.–07:59 a.m. | 08:00 a.m.–10:40 p.m. |
| R-BRA02 | 10:01 p.m.–05:59 a.m. | 06:00 a.m.–10:00 p.m. |
| R-BRA03 | 10:01 p.m.–07:59 a.m. | 08:00 a.m.–10:00 p.m. |
| R-BRA04 | 09:16 p.m.–08:59 a.m. | 09:00 a.m.–09:15 p.m. |
| R-BRA05 | 11:59 p.m.–07:34 a.m. | 07:35 a.m.–11:58 p.m. |
| R-BRA06 | 09:51 p.m.–05:29 a.m. | 05:30 a.m.–09:50 p.m. |
| R-BRA07 | 08:51 p.m.–05:59 a.m. | 06:00 a.m.–08:50 p.m. |

We used the Parkes error grid (PEG) [56] to evaluate errors in the measurement of predicted glucose oscillation provided by Tesseratus. PEG was adopted because it is compliant with ISO15197 [57] and separates T1D individuals from individuals with type 2 DM. Moreover, PEG had its technical issues revised by Pfutzner et al. 2013 [54], who established exact borders for the performance zones for glucose measurements and supported the accuracy definition for glucose monitors. The zones are categorized as A, B, C, D and E. The associate meaning for a measurement of being in a zone is: zone A—clinically accurate measurements, no effect on clinical action; zone B—altered clinical action, little or no effect on clinical outcome; zone C—altered clinical action, likely to affect clinical outcome; zone D—altered clinical action, could have significant clinical risk; zone E—altered clinical action, could have dangerous consequences.

PEG was used considering the predicted **versus** measured values, and the success metric was simple: the more prediction points that fall within zones A and B, the better the model. Table 7 details the PEG for the daytime window considering each prediction horizon (PH) (from 15 min to 4 h). It can be depicted that 95.1% of measurements, on average, fall in zones A and B. The same result is presented in Figure 7, showing a total of 3804 predictions. Zones A and B are the closest ones to the diagonal line of the plot.

**Table 7.** Results of PEG for Tesseratus (3804 predictions plotted)—daytime period.

| PH | A (%) | B (%) | C (%) | D (%) | E (%) | A + B (%) |
|---|---|---|---|---|---|---|
| 15 min | 93 | 6.8 | 0.2 | 0 | 0 | 99.8 |
| 30 min | 76.9 | 22.2 | 0.8 | 0 | 0 | 99.1 |
| 60 min | 56.4 | 41.5 | 2 | 0.1 | 0 | 97.9 |
| 90 min | 50.1 | 44.9 | 4.7 | 0.3 | 0 | 95 |
| 120 min | 50.3 | 41.8 | 7.4 | 0.5 | 0 | 92.1 |
| 180 min | 46 | 44 | 8.9 | 0.9 | 0.2 | 90 |
| 240 min | 49.6 | 40.4 | 9.7 | 0.3 | 0 | 90 |
| **Average** | 62.3 | 32.8 | 4.4 | 0.5 | 0 | **95.1** |

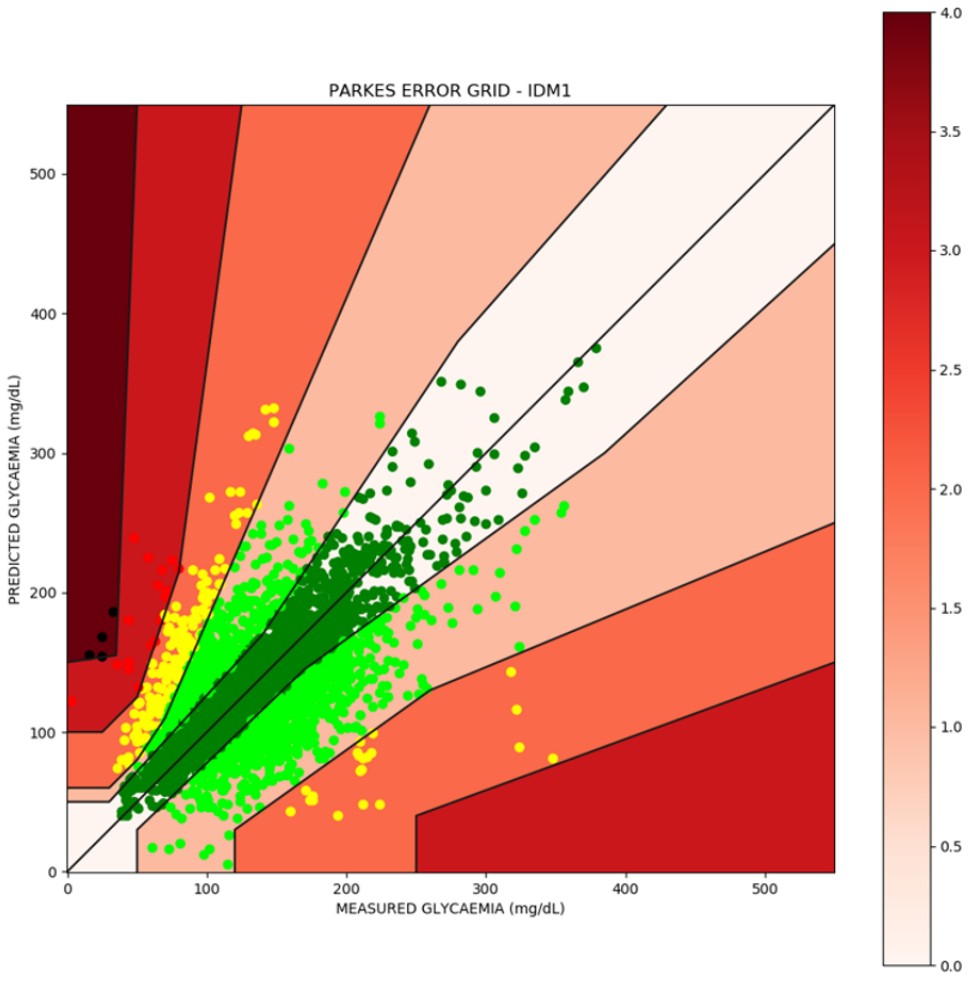

**Figure 7.** PEG (daytime period) for seven T1D individuals. Green dots lay at zones A and B (95.1% on average): 3804 predictions.

For the night-time window, Table 8 details the PEG of each PH (from 15 min to 8 h). It can be depicted that 93.7% of measurements, on average, fall in zones A and B. The same result is presented in Figure 8, showing a total of 1400 predictions. It is important to observe that, for a PH of 480 min (eight hours), the Tesseratus model presented a PEG of 95% falling between zones A and B.

**Table 8.** Results of PEG for Tesseratus (1400 predictions plotted)—night period.

| PH | A (%) | B (%) | C (%) | D (%) | E (%) | A + B (%) |
|---|---|---|---|---|---|---|
| 15 min | 86 | 12.4 | 1.7 | 0 | 0 | 98.4 |
| 30 min | 72.2 | 26.9 | 0.9 | 0 | 0 | 99.1 |
| 60 min | 57.1 | 41.8 | 1.1 | 0 | 0 | 98.9 |
| 90 min | 50 | 44.4 | 5.6 | 0 | 0 | 94.4 |
| 120 min | 47.7 | 46.7 | 5.6 | 0 | 0 | 94.4 |
| 180 min | 45.3 | 44.7 | 8.6 | 0.7 | 0.6 | 90 |
| 240 min | 48.8 | 42.2 | 8.4 | 0.6 | 0 | 91 |
| 300 min | 46.2 | 44.8 | 8 | 1.2 | 0 | 91 |
| 360 min | 54.2 | 37.8 | 6.5 | 1.5 | 0 | 92 |
| 420 min | 52.3 | 39.7 | 6.7 | 1.3 | 0 | 92 |
| 480 min | 62 | 33 | 4.5 | 0.5 | 0 | 95 |
| **Average** | 55.8 | 37.9 | 5.6 | 0.7 | 0 | **93.7** |

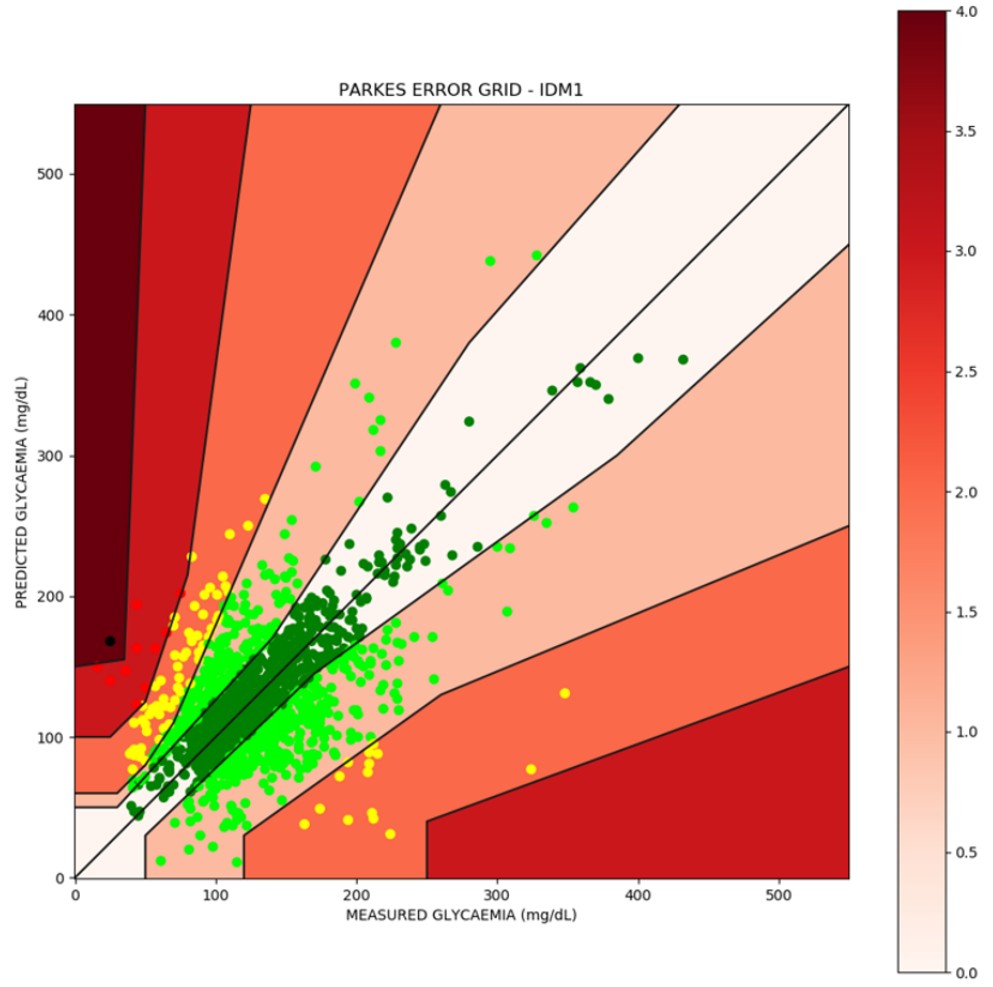

**Figure 8.** Parkes error grid (night period) for seven T1D individuals. Green dots lay at zones A and B (93.7% on average): 1400 predictions.

Among other well-known error evaluation metrics, such as mean absolute percentage error (MAPE), mean absolute error (MAE) and root mean square error (RMSE), we considered MAE and MAPE the most relevant to evaluate the success of our model, because T1D individuals need predictions as close to real measurements as possible. Thus, the lower the MAE and MAPE values, the better the model is in relation to the prediction.

In fact, such metrics are calculated considering absolute values for the difference (measured value minus predicted glucose concentration). Therefore, we evaluate MAE and MAPE considering such errors for all PHs, and organized them for daytime and nightt-ime in Tables 9 and 10, respectively. It is important to note that the daytime window runs from 15 to 240 min and night-time window from 15 to 480 min.

**Table 9.** Mean Absolut Error (MAE)—daytime and nighttime comparison of PH concerning seven T1D individuals. General average MAE at daytime and nighttime for Tesseratus.

| PH | Daytime (avg in mg/dL) | Night-Time (avg in mg/dL) |
|---|---|---|
| 15 min | 9.18 | 9.27 |
| 30 min | 16.97 | 16.09 |
| 60 min | 26.41 | 26.26 |
| 90 min | 30.24 | 28.09 |
| 120 min | 30.54 | 28.3 |
| 180 min | 33.61 | 32.68 |
| 240 min | 25.01 | 34.49 |
| 300 min | – | 35.16 |
| 360 min | – | 34.97 |
| 420 min | – | 33.96 |
| 480 min | – | 26.37 |
| **Tesseratus** | 24.56 | 27.77 |

**Table 10.** Mean Absolut Percentage Error (MAPE)—daytime and nighttime comparison of PH concerning seven T1D individuals. General average MAPE at daytime and nighttime for Tesseratus.

| PH | Daytime (%) | Night-Time (%) |
|---|---|---|
| 15 min | 8 | 7.84 |
| 30 min | 13.32 | 14.43 |
| 60 min | 20.52 | 22.78 |
| 90 min | 24.98 | 28.11 |
| 120 min | 30.79 | 28.04 |
| 180 min | 33.26 | 31.19 |
| 240 min | 37.82 | 30.57 |
| 300 min | – | 23.96 |
| 360 min | – | 36.49 |
| 420 min | – | 32 |
| 480 min | – | 25.33 |
| **Tesseratus** | 22.01 | 25.51 |

## 6. Discussion

Both results linked to PEG analysis and MAE are pioneers, due to the long-term prediction horizons, e.g, seven or eight hours for night-time. To the best of our knowledge,

there is no literature presenting these prediction horizons. However, we compared Tesseratus with other models that address a shorter PH. In fact, whenever we consider PEG, we can cite [37,58] for a PH of one hour. On one hand, the study by Munoz-Organero [37] considered nine real individuals and obtained a PEG of 87.22% falling in zones A + B, while the study by Foss-Freitas et al. [58] considered 21 real individuals and obtained a Clarke error grid (CEG) of 94% falling in zones A and B. It is worth noting that both PEG and CEG are comparable error grids [59]. Tesseratus outperformed both for the same PH, achieving a PEG of 97.9% falling in zones A and B for daytime, and 98.9% at night-time.

Whenever we consider MAE as the error, we can cite [60], who obtained an MAE equal to 51.3 mg/dL for a PH of 60 min, whereas Tesseratus reached 26.41 mg/dL, both at daytime. It is worth noting that the smaller the MAE, the better the values. Finally, if the considered error is MAPE, Foss-Freitas et al., 2019 [58] presented interesting results for a PH ranging from 30 to 360 min. A comparison with Tesseratus is presented in Table 11 considering night-time PH. Observe that Tesseratus outperforms them for a PH of 120, 180 and 360 min, the last one being the highest PH that they achieve. Nevertheless, they outperform Tesseratus for short PHs such as 30 and 60 min. We advocate that this is not an issue, since the need for long-term prediction is paramount for night-time.

**Table 11.** MAPE at nighttime comparison between Foss-Freitas et al., 2019 [58] and Tesseratus—21 days of training.

| PH | Foss-Freitas et al., 2019 [58] (%) | Tesseratus (%) |
| :---: | :---: | :---: |
| 30 min | 7 | 14.43 |
| 60 min | 16.8 | 22.78 |
| 120 min | 32.7 | **28.04** |
| 180 min | 45 | **31.19** |
| 360 min | 44.2 | **36.49** |
| 420 min | – | **32** |
| 480 min | – | **25.33** |

In addition to the results achieved, it is worth mentioning the main contributions of the Tesseratus model in a summarized way: (1) a combination of different techniques in the same model working in an orchestrated way, from mathematics to ML (supervised and reinforcement learning), represented by agents; (2) continuous learning for applicability in real individuals with T1D; (3) delegating the task of a continuous self-adjustment process about prediction errors to the agents; (4) Tesseratus works independently of sex and age; (5) it was tested with real individuals; (6) applicability in technology-based healthcare; (7) prediction of glucose oscillation of up to eight hours, depending on the individual's lifestyle, with an acceptable absolute error; (8) support for personalized recommendations on macronutrients, insulin and physical exercise, based on ML models, and not just fixed rules.

All of the aforementioned contributions indicate the feasibility of Tesseratus to be embedded as the underlying model to: an open source artificial pancreas system [61]; a commercial artificial pancreas (closed-loop) [62]; insulin pumps [63,64]; and a recommendation system, to cite a few. It is our claim that it must facilitate the daily life of T1D individuals, automating most of the individual T1D's 180 extra daily health-related decisions [65].

## 7. Conclusions

The paper presented Tesseratus, a hybrid model that adopts a multi-agent approach to address the problem of predicting glucose concentration for T1D individuals. Tesseratus has reactive and intelligent agents, where the reactive agents act as sensors and monitors, and intelligent agents act as an oracle or as an apprentice. The oracle knowledge is provided by a mathematical model for the HGIRS and it transfers knowledge to the apprentice, which

also learn from data provided by sensors and monitors. Tesseratus uses a dual continuous learning model that can mitigate errors between the predicted and continuously measured values, in addition to the ODE's own input parameters. The combination of techniques is advantageous, as it consists of models with complementary functions, resulting in a cohesive, well-adjusted model capable of generalization.

Tesseratus was validated with seven real T1D Brazilian individuals that provided their data collected for up to 21 days: the seven initial days were used to personalize the model and the following days were used to validate the model capacity of predicting glucose oscillation for a PH that ranges from 15 min to 4 h during daytime and from 15 min to 8 h at night-time. As the evaluation of the MAE indicates, Tesseratus is able to predict glucose oscillation with an accuracy equal to or less than 30 mg/dL (1.7 mmol/L). As future work, it is our intention to continuously improve the performance of the Tesseratus while reducing the error of the predicted value by removing some specific restrictions found in the literature: (a) constant values of parameters from mathematical models for prediction calculations; (b) agent's support for continuous learning and ODE parameter values correction; (c) barriers to combining different prediction models, using active learning, reusing, combining and adapting knowledge from different agents [66].

Some limitations need to be addressed in future work, such as: (1) it does not support pregnant women and type 2 individuals with diabetes mellitus; (2) for now, it only supports four types of insulin analogues: aspart, lispro, glargine and degludec; (3) the dataset is still small, performed with historical data for only seven real individuals; (4) few insulin sensors or pumps were tested during the research, according to Table 4; (5) it is necessary to test with new variables: stress, hormonal effects, blood oxygen and heart rate; (6) it is necessary to analyze blood glucose and insulin concentration associated with comorbidities.

**Author Contributions:** Conceptualization, J.P.A.P., A.A.F.B., J.d.S.B. and M.L.C.C.-G.; methodology, J.P.A.P. and A.A.F.B.; software, J.P.A.P.; validation, J.P.A.P., A.A.F.B. and J.d.S.B.; formal analysis, J.P.A.P. and J.d.S.B.; investigation, J.P.A.P.; resources, J.P.A.P. and A.A.F.B.; data curation, J.P.A.P.; writing—original draft preparation, J.P.A.P.; writing—review and editing, J.P.A.P., A.A.F.B., J.d.S.B. and M.L.C.C.-G.; visualization, J.P.A.P.; supervision, A.A.F.B.; project administration, J.P.A.P. and A.A.F.B. All authors have read and agreed to the published version of the manuscript.

**Funding:** This research received no external funding.

**Institutional Review Board Statement:** The study was conducted in accordance with the Declaration of Helsinki, and approved by the Ethics Committee of the University Hospital of São Paulo (HU-USP) (protocol code 1819188—approved on 24 September 2021), for studies involving humans.

**Informed Consent Statement:** Informed consent was obtained from all subjects involved in the study (Free and Informed Consent Term).

**Data Availability Statement:** Not applicable.

**Conflicts of Interest:** The authors declare no conflict of interest.

## Abbreviations

The following abbreviations are used in this manuscript:

| | |
|---|---|
| AC | Abdominal Circumference |
| AP | Artificial Pancreas |
| BG | Blood Glucose |
| BM | Body Mass |
| CGM | Continuous Glucose Monitoring |
| CNS | Central Nervous System |
| DM | Diabetes Mellitus |
| EGA | Error Grid Analysis |
| eGDR | estimated Glucose Disposal Rate |
| HBA1C | Glycated Hemoglobin |

|       |                                        |
|-------|----------------------------------------|
| HGIRS | Human Glucose–Insulin Regulatory System |
| HGP   | Hepatic Glucose Production             |
| IDF   | International Diabetes Federation       |
| IT1D  | Individuals with T1D                    |
| MAE   | Mean Absolute Error                     |
| MAS   | Multi-Agent System                      |
| ML    | Machine Learning                        |
| ODE   | Ordinary Differential Equations         |
| PEG   | Parkes Error Grid                       |
| RMSE  | Root Mean Square Error                  |
| PH    | Prediction Horizon                      |
| SBD   | Brazilian Society of Diabetes           |
| T1D   | Type 1 Diabetes Mellitus                |
| TEX   | Time of Exercise                        |
| VO2   | Volume of Oxygen                        |

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
