# Peer review of "A Multi-Agent Approach Used to Predict Long-Term Glucose Oscillation in Individuals with Type 1 Diabetes"

_applsci, doi:10.3390/app12199641_

Round 1

Reviewer 1 Report

SUMMARY OF THE PAPER: This work presents a platform based on agents, machine learning and mathematical modeling to predict the glucose oscillation of T1D individuals up to 8 hours.

FORMATTING REVIEW: Although it is a demanding article for the reader due to the amount of information it provides, it is very well structured and written. The figures and tables are clear and provide relevant information to support the text.

CONTENT REVIEW: I have no objection to the methodology and results of the research. However, it seems to me that the formalization of the platform around the agent architectural paradigm is not properly justified: Why do the authors choose the multi-agent architectural style? Why doesn't Tesseratus follow another type of architectural paradigm, such as microservices?

Currently, microservices are booming in the cloud, but they remain to be adopted and adapted in other domains. On the other hand, the agent paradigm has not been so widely accepted, but it has been widely adopted in academia. In addition, the use of multi-agent systems is still being promoted in eHealth and other areas (IEEE Std 2660.1 - 2020 Standard ("IEEE Recommended Practice for Industrial Agents," 2021).

As with the microservices-based paradigm, multi-agent systems can solve problems that are difficult to solve with a monolithic approach. But, unlike the microservices-based paradigm, multi-agent systems natively provide functional mechanisms for building applications with distributed intelligence.

I believe that the authors should justify their architectural style taking into account these aspects and/or others that they may consider appropriate.

Finally, the article states that Tesseratus was implemented in Python, but no information is given about its development. Is it FIPA compliant (does it use ACL?)? Was some kind of agent framework (such as JADE) used to develop it?

Author Response

Thank you so much about comments. I added information about CSP (Cloud Service Provider - Azure) that I implemented the Tesseratus, based on serverless, event-driven and microservices. We used MAS to keep the self-correction continuously, and developed using Python, Spade-BDI, and to keep the competition between different agents (ML and mathematical, or among others). I worked on AWS, and I work ate MSFT Azure, for this reason make sense add this information about the implementation aspects.

Reviewer 2 Report

 The authors propose an approach based on a multi-agent system and machine learning model to predict glucose oscillation in Individuals with T1D. They show some performance metrics and compare the performance of their proposals with other algorithms in the literature. However, I have some recommendations and concerns about the manuscript:

  1. Add the results in the abstract.
  2. In the last paragraph in the Introduction section, please mention section 6.
  3. You numbered the equation in section 2.1.1 (The glucose equation).
  4. You numbered the equations on page 5 (from y1 to y5) and clarify the second group of equations on page 5 (from y1 to y5) are the same. Two equations “y5”? 
  5. In the Related Work section, please explain some references from [37] to [46].
  6. Change table for Table and figure for Figure or Fig., according to the guide for authors.
  7. What is the difference between Fig. 2 and Fig. 3?
  8. Explain in detail in Fig. 5 and add units for the y-axis.
  9. For numbers from 0 to 10, use letters (one, two, three, and so far). Use numbers (150, 300, and so far) for values greater than ten.
  10. What version of Python do the authors use?
  11. In all tables, highlight only the best results.
  12. Throughout the manuscript, check for typo errors and spelling.

Author Response

Thank you for your comments! They will be addressed in the following:

  1. Add the results in the abstract. - Results added in the abstract.
  2. In the last paragraph in the Introduction section, please mention section 6. - Mentioned the section 6 (line 59).
  3. You numbered the equation in section 2.1.1 (The glucose equation). - Done: now it is equation 2
  4. You numbered the equations on page 5 (from y1 to y5) and clarify the second group of equations on page 5 (from y1 to y5) are the same. Two equations “y5”? - The paragraphs that described Figures 1 and 2 were rewritten and the polynomials were described as piecewise functions.
  5. In the Related Work section, please explain some references from [37] to [46]. In section 3, more references with results with real individuals and respective techniques were added: [37], [39] and [46].
  6. Change table for Table and figure for Figure or Fig., according to the guide for authors.  Fixed all (changed table by Table and figure to Figure).
  7. What is the difference between Fig. 2 and Fig. 3? Thank you so much. We did not realize it. We replaced Figure 3 by the correct one. Now, it is in conformance with the explanation flow from lines 227 to 253.
  8. Explain in detail in Fig. 5 and add units for the y-axis. In the subsection 4.3 was detailed the diagram and units about the Figure 5. 
  9. For numbers from 0 to 10, use letters (one, two, three, and so far). Use numbers (150, 300, and so far) for values greater than ten. Fixed all.
  10. What version of Python do the authors use? version 3.8 of Python was added.
  11. In all tables, highlight only the best results. Done. Highlighted in Table 11, Tables 7 and 8 (only Tesseratus result).
  12. Throughout the manuscript, check for typo errors and spelling. I fixed: for example: degreea by degrees, TD1 by T1D and others.

Reviewer 3 Report

As a pediatric endocrinologist, I am not qualified to analise in depth the proposed model, in terms of mathematic adequacy or soundness, but in terms of design, considering that type 1 diabetes, the main objective for the proposed model, is a great concern for young patients (infants, children and adolescents), the largest group of type 1 patients in our country, that characteristically have much more glycemic fluctuations, when compared with adult patients, as the small sample used for collecting your "real life" data, in my opinion is the greatest weakness of your work. This was not even mentioned on your conclusions, when admitted limitations related only for not being suitable for pregnant or type 2 diabetic patients, and minor flaws, that would not compromise the core of the proposition, intended for prediction of a wider time horizon, mainly sleep time, that is always a challenge for Type 1 diabetic patients, specially on pediatric age.

Author Response

Thank you so much about all comments. All information was relevant, mainly about hormones, different source/data and assumptions.

1 - ABSTRACT: Assumption of great impact of Tesseratus with apparently few consistent preliminary data.

The idea of the Tesseratus model is to customize each prediction model for each individual with T1D. Thus, the number of prediction calls, with these seven real individuals, reached 3.804 during the day and 1,400 at night, with acceptable errors (MAE <= 28 mg/dL) with prediction horizons of up to 8 hours.

2 - SECTION 2: Oversimplification - ignores important hormones such as cortisol and growth hormone, that exerts direct and indirect effects on glucose metabolism.

We worked with the minimal model to prove that it is possible to predict glycemic oscillation, with the main hormones (insulin and glucagon) and other data such as glucose, carbohydrates, exercise and exogenous insulin types.

3 - Seems to me much more assumptions (subsection 4.2):

Here is a prediction technique in AI/ML, where I can analyze all historical data, or only part of it. In this case, the hypotheses of bolus insulin action (5 hours) and carbohydrate metabolism in the human body (2 hours) were used.

4 - Her profile, since its a female volunteer.

Changed from His to Her.

5 - different sources of data, few patients and different insulin regimens

The idea was to demonstrate the versatility of the Tesseratus model with different data sources, with individuals of different age ranges, in addition to flash sensors and commercial insulin pumps, in addition to different types of insulin. It was possible to demonstrate the ability to create a personalized model for each individual with T1D.